# Arterial Blood Gas Analysis for Survival Prediction in Pediatric Patients with Out-of-Hospital Cardiac Arrest

**DOI:** 10.3390/jpm13071061

**Published:** 2023-06-28

**Authors:** Sang-Hwan Lee, Hyungoo Shin, Yongil Cho, Jaehoon Oh, Hyuk-Joong Choi

**Affiliations:** Department of Emergency Medicine, College of Medicine, Hanyang University, Seoul 04763, Republic of Korea; sangwhan123@hanyang.ac.kr (S.-H.L.); ojjai@hanmail.net (J.O.); ardoc@hanyang.ac.kr (H.-J.C.)

**Keywords:** out-of-hospital cardiac arrest, return of spontaneous circulation, blood gas analysis, pediatric

## Abstract

Arterial blood gas analysis (ABGA) is one of the few tests performed during cardiopulmonary resuscitation (CPR). There have been some studies on the prediction of survival outcomes in adult out-of-hospital cardiac arrest (OHCA) patients during CPR using ABGA results. However, in pediatric OHCA patients, the prognosis of survival outcome based on ABGA results during CPR remains unclear. We retrospectively analyzed prospectively collected data from the Korean Cardiac Arrest Resuscitation Consortium (KoCARC) registry, a multicenter OHCA registry of Republic of Korea. We analyzed 108 pediatric (age < 19 years) OHCA patients between October 2015 and June 2022. Using multivariable logistic regression, an adjusted odds ratio (aOR) was obtained to validate the ABGA results of survival to hospital admission and survival to discharge. The variables associated with survival to hospital admission were non-comorbidities (aOR 3.03, 95% confidence interval (CI) 1.22–7.53, *p* = 0.017) and PaO_2_ > 45.750 mmHg (aOR 2.69, 95% CI 1.13–6.42, *p* = 0.026). There was no variable that was statistically significant association with survival to discharge. PaO_2_ > 47.750 mmHg and non-comorbidities may serve as an independent prognostic factor for survival to hospital admission in pediatric OHCA patients. However, the number of cases analyzed in our study was relatively small, and there have been few studies investigating the association between ABGA results during CPR and the survival outcome of pediatric OHCA patients. Therefore, further large-scale studies are needed.

## 1. Introduction

Approximately 11.4% of pediatric out-of-hospital cardiac arrest (OHCA) patients survive hospital discharge in the United States [1]. According to one study, the rate of survival to discharge following OHCA has increased annually for adults but has not improved for pediatric OHCA patients [2]. The survival to discharge rate in pediatric OHCA patients was approximately 11.4%–13.7% in Republic of Korea. However, in the case of pediatric in-hospital cardiac arrest patients, it was over 40% [3]. Since the life expectancy of children is longer than that of adults, the individual and societal consequences of their deaths are significant. Many studies are needed to improve the survival rate of OHCA in pediatric patients.

Cardiac arrest and the return of spontaneous circulation result in systemic ischemia and reperfusion syndrome. This results in metabolic acidosis and severe hypercapnia, which lead to the accumulation of carbon dioxide, hydrogen ions, and lactic acid [4,5]. Considering the low survival rate of out-of-hospital cardiac arrest patients, research on patient prognosis is very important, both at an individual and societal level. Research has been conducted on biochemical markers to evaluate the prognosis of patients after cardiac arrest and spontaneous circulation, due to the physiological changes that occur during post-arrest recovery. However, there is a few studies specifically focusing on pediatric patients compared to adults. Furthermore, there is currently no factor that has demonstrated sufficient accuracy in predicting the outcome of pediatric post-arrest survival [6]. Among the biochemical markers of pediatric cardiac arrest, researches have suggested that neuron-specific enolase (NSE), serum lactate and pH measured within the first 24 hours after arrest, and the delta neutrophil index are associated with patient survival and neurological prognosis [6,7,8]. However, the specific cutoff values for each biological marker are not well-established, and there is a limited amount of research on biological markers during cardiac arrest.

Arterial blood gas analysis (ABGA) may provide useful diagnostic and prognostic information about patients [9]. Moreover, the results of point-of-care testing (POCT) can be quickly obtained. There have been some studies on the prediction of the outcome of adult OHCA patients during cardiopulmonary resuscitation (CPR) using ABGA [10,11,12,13,14]. However, there are few studies on the efficacy of ABGA for the outcome of pediatric OHCA patients. Actually, ABGA is widely used in the clinical field during CPR due to the relatively simple and quick results it provides.

The objective of our study was to investigate the association between initial ABGA results and the survival outcome of pediatric OHCA patients.

## 2. Materials and Methods

### 2.1. Study Design and Variables

We retrospectively analyzed data that was prospectively collected from the Korean Cardiac Arrest Research Consortium (KoCARC) registry. This study included pediatric patients with OHCA included in the KoCARC registry between 1 October 2015, and 30 June 2022. KoCARC is a multicenter collaborative research network of hospitals in the Republic of Korea [15]. The KoCARC registry was registered at clinicaltrials.gov (accessed on 17 April 2023) as protocol NCT 03222999. In this study, OHCA patients younger than 19 years (age range defined as pediatric by the Korean Pediatric Society) transported to an emergency department (ED) by any method and with any etiology of cardiac arrest were included in the registry.

We analyzed several variables from the KoCARC registry: (1) patient information and outcome, including age, sex, ED direct visit or transfer from another hospital, result of ED resuscitation, result of ED disposition, and result of disposition after hospital discharge; (2) community resuscitation, including witness and bystander CPR; (3) emergency medical service (EMS) resuscitation, including first EMS electrocardiography (ECG) rhythm, advanced airway, and return of spontaneous circulation (ROSC) at scene; (4) hospital resuscitation, including initial ECG rhythm on arrival at ED and ABGA result; and (5) pediatric resuscitation, including pediatric CPC at hospital discharge and past medical history.

In this study, the first monitored ECG was defined as the first EMS ECG rhythm. When the EMS ECG rhythm was missing or if the patient was not transported through EMS in the prehospital state, the first cardiac arrest ECG rhythm at the ED was defined as the first monitored ECG.

### 2.2. Participants

This study comprised pediatric patients with OHCA included in the KoCARC registry between 1 October 2015, and 30 June 2022. We excluded patients without CPR results, patients with cardiac arrest of traumatic etiology, patients with missing pH values, and patients with ROSC at the scene that could significantly impact the initial ABGA. And patients transferred from other hospitals were excluded from this study because the managements performed at other hospitals could impact the ABGA.

### 2.3. Outcome Variables

Outcomes were analyzed with the above variables for (1) survival to hospital admission and (2) survival to discharge. Survival to hospital admission was defined as the maintenance of ROSC until arrival at the ED and the transfer of care to medical staff at the receiving hospital. This definition corresponds to the ‘survival event’, which is a core outcome of the 2015 Utstein template [16]. Survival to discharge was defined as the patient being alive at the point of hospital discharge.

### 2.4. Statistical Analysis

The baseline characteristics and variables are presented with descriptive statistics. Categorical variables are presented as numbers and percentages using the chi-square or Fisher’s exact test. The normality test of continuous variables was performed using the Shapiro-Wilk test. Continuous variables with normality are presented as the mean and standard deviation (mean ± standard deviation), and continuous variables with non-normality are presented as the median and interquartile range (median [25th percentile–75th percentile]). A student’s t test or Wilcoxon rank sum test was performed depending on the results of the normality test.

To identify independent indicators of prognosis, we describe survival to hospital admission and survival to discharge as adjusted odds ratios (aOR) and 95% confidence intervals (CI) in OHCA patients using multivariable logistic regression analysis, integrating covariates (variables with a *p*-value < 0.05) from the results of our univariable logistic regression. To identify the optimal cutoff value of the ABGA result for predicting survival outcome, we used receiver operating characteristic (ROC) curves and the closest-to-(0, 1) method [17]. Statistical significance was determined when the *p*-value was less than 0.05 on a two-sided test. All statistical analyses were performed using R version 4.2.2 (www.R-project.org, accessed on 1 April 2023).

## 3. Results

There were a total of 429 pediatric patients younger than 19 years of age whose data were entered into the KoCARC registry between 1 October 2015, and 30 June 2022. Among these, the following patients were excluded based on the exclusion criteria: patients without results of ED resuscitation (*n* = 5), patients with spontaneous circulation before hospitalization (*n* = 53), patients transferred from other hospitals (*n* = 12), patients transferred to another hospital (*n* = 23), patients with trauma as the cause of cardiac arrest (*n* = 136), and patients without pH values in the ABGA results (*n* = 92). Therefore, a total of 108 patients were analyzed (Figure 1).

### 3.1. ABGA Result for Prediction of Survival to Hospital Admission

The demographic characteristics of the 108 pediatric OHCA patients are presented in Table 1. Of these, 33 patients (30.6%) were hospitalized with successful ROSC after OHCA, and 75 patients (69.4%) died in the ED. There was a significant difference in comorbidities (survival 9 (27.3%) vs. non-survival 40 (53.3%)). However, there were no significant differences in other variables, including ABGA results.

A logistic regression analysis was performed using the variables in Table 2. The results of ABGA were converted from continuous variables to categorical variables to identify the optimal cutoff value. The cutoff values for each variable were 6.795 (area under the curve (AUC), 0.623; 95% CI, 0.509–0.737) for pH, 92.75 mmHg (AUC, 0.597; 95% CI, 0.481–0.712) for PaCO_2_, 45.750 mmHg (AUC, 0.613; 95% CI, 0.489–0.737) for PaO_2_, and 16.6 mmol/L (AUC, 0.483; 95% CI, 0.334–0.631) for lactate (Appendix A).

In the univariable logistic regression analysis (Table 2(A)), the variables associated with survival to hospital admission were PaO_2_ > 45.750 mmHg (odds ratio 2.71; 95% CI 1.17–6.30, *p* = 0.020) and non-comorbidities (odds ratio 3.05; 95% CI 1.25–7.42, *p* = 0.014). In the multivariable logistic regression analysis (Table 2(B)), the variables associated with survival to hospital admission were comorbidities (aOR 3.03; 95% CI 1.22–7.53, *p* = 0.017) and PaO_2_ > 45.750 mmHg (aOR 2.69; 95% CI 1.13–6.42, *p* = 0.026).

### 3.2. ABGA Result for Prediction of Survival to Discharge

The demographic characteristics of the 32 pediatric OHCA patients who survived hospital admission are presented in Table 3. Of these, 12 patients (37.5%) survived to discharge, and 20 patients (62.5%) died after hospital admission. There was a significant difference in pH (survival 6.87 ± 0.15 mmHg vs. non-survival 6.73 ± 0.18 mmHg, *p* = 0.030).

A univariable logistic regression analysis was performed using the variables in Table 3. The results of ABGA were converted from continuous variables to categorical variables. The cutoff values for each variable were 6.775 (AUC, 0.733; 95% CI, 0.553–0.914) for pH, 66.5 mmHg (AUC, 0.625; 95% CI, 0.391–0.859) for PaCO_2_, 47.85 mmHg (AUC, 0.510; 95% CI, 0.293–0.728) for PaO_2_, and 14.865 mmol/L (AUC, 0.625; 95% CI, 0.379–0.871) for lactate (Appendix A).

In the univariable logistic regression analysis (Table 4(A)), the variables associated with survival to discharge were pH > 6.775 (odds ratio 7.5; 95% CI 1.29–43.69, *p* = 0.025) and PaCO_2_ ≤ 66.500 mmHg (odds ratio 6.8; 95% CI 1.23–37.50, *p* = 0.028). In the multivariable logistic regression analysis, there was no statistically significant variable (Table 4(B)).

## 4. Discussion

This study investigated the efficacy of initial ABGA results for predicting survival outcomes in pediatric OHCA patients. The variables associated with survival to hospital admission were comorbidities and PaO_2_ > 45.750 mmHg. There was no statistically significant variable associated with survival to discharge.

This study collected data from hospitals in almost all regions of Republic of Korea and investigated a relatively large number of pediatric OHCA patients, which is an uncommon condition in pediatric settings. A total of 429 patients were enrolled in our study, with 321 excluded based on various exclusion criteria, resulting in an analysis of 108 patients. Although a number of patients were excluded, comparing the survival to discharge rates of 11.4%–13.7% reported in previous studies on pediatric cardiac arrest epidemiology in Republic Korea to the 11.1% (12 out of 108) survival to discharge rate in our study shows no substantial difference [2]. Considering only the survival to discharge metric, it could be suggested that our data to some extent represent Republic of Korea.

During CPR, the available diagnostic examinations are limited. According to the American Heart Association’s 2020 guidelines for adult basic life support (BLS) and advanced cardiovascular life support (ACLS), the use of ultrasound by an experienced sonographer and end-tidal CO_2_ measurement are weakly recommended during cardiac arrest treatment, while routine measurement of arterial blood gases during CPR is considered uncertain [6,18]. However, there is ongoing research on the relationship between the results of blood gas analysis and survival outcomes [10,11,14,19,20,21]. In contrast, the pediatric BLS and ACLS guidelines provide a weak recommendation for echocardiography and ETCO_2_, but they do not mention blood gas analysis or specific mention of PaO_2_ and PaCO_2_ during cardiac arrest. A few studies have been conducted on pediatric OHCA patients. The prognosis based on ABGA results during CPR remains unclear.

One study investigated the association between the initial serum pH value and the outcomes of pediatric out-of-hospital cardiac arrest [22]. This study concludes that the initial ABGA result may be a useful indicator of the survival outcome of pediatric OHCA patients. However, this study did not analyze ABGA results other than the pH value. Additionally, there was heterogeneity in the enrolled patients. Unlike our study, this study included patients with cardiac arrest caused by non-trauma, drug overdose, drowning, and suffocation, among others. However, in our study, there was no significant association between pH and any survival outcome. But, pH may be a marker for the severity of hypoperfusion. Although the association between survival to discharge and pH had a p-value of 0.144, which was higher than 0.05, the adjusted odds ratio was relatively high at 4.15. We also do not believe that there is no association between pH and survival outcome at all. As there was a relation between pH and survival outcome in a previous large-scale adult study, it is thought that further large-scale studies in pediatric patients are needed.

Some studies have shown that a higher PaO_2_ during CPR is associated with ROSC [11,14,19]. However, no human studies have directly compared the level of inspired oxygen concentration during CPR. The physiologic mechanisms between high PaO_2_ results and higher rates of ROSC are not uncertain, but the clinical implications of our study findings suggest the following: PaO_2_ and PaCO_2_ values are dependent on cardiac output and ventilation, so patients with high PaO_2_ might receive high-quality CPR. In ventilation, prehospital advanced airway was not a significant survival outcome factor in our study. Additionally, one previous study reported that prehospital advanced airway was not associated with survival outcomes [23]. Therefore, the effect of chest compression rather than ventilation is thought to have had a greater effect on the patient’s outcome. In other words, while maintaining a proper gas exchange in the airway and alveoli is important, it is crucial to prioritize high-quality CPR for gas exchange between the alveoli and pulmonary artery, as well as for systemic circulation. However, after arriving at the hospital, airway management could not be checked in our data, and respiratory arrest accounted for a greater proportion of pediatric cardiac arrest than adult cardiac arrest. The importance of ventilation should not be overlooked.

Cardiac arrest causes systematic tissue hypoxia, resulting in the production of high lactate levels during anaerobic metabolism [24]. However, our study did not find that lactate levels during CPR were associated with survival. Topjian and colleagues reported that lactate levels after successful resuscitation from pediatric cardiac arrest are associated with increased mortality [7]. However, there is no study on lactate levels and survival outcomes during pediatric CPR. The association of lactate value during adult cardiac arrest with survival outcomes is controversial [25,26].

In our study, comorbidities were associated with survival after hospital admission. Previous studies reported that comorbidities are less associated with survival to discharge [7,27]. However, there was a significant difference only in survival after hospital admission in our study. The exact reason for this outcome is unclear. However, we thought as follows: even if the comorbidity had been found at a young age, the effect on the body would have been severe. When cardiac arrest occurs, resuscitation may be difficult due to severe comorbidities. Pediatric patients with severe comorbidities expire in the ED, so it is thought that there will be no significant difference in survival after hospital admission. However, since our data lacked information on the severity and clear diagnosis of comorbidities, interpretation must be performed with caution.

The patient’s prognosis can be evaluated through the ABGA. However, relying solely on ABGA to assess patient outcomes may lead to overestimation. While ABGA provides information on the patient’s condition, subsequent medical interventions, such as high-quality CPR and post-cardiac arrest syndrome care by healthcare professionals, can also influence survival outcomes. While the initial patient condition has a significant impact on the prognosis, healthcare professionals should exert efforts in their management and should not make hasty judgments regarding the termination of resuscitation. Our study described the aOR of variables for survival outcome, but it does not mean the termination of resuscitation. When terminating resuscitation, several factors, such as prehospital stages, hospital stages, comorbidities, and the emotions of the patient’s guardian, should be considered.

There were notable limitations to our study. First, several factors that could affect ABGA could not be investigated. Drugs such as sodium bicarbonate, prehospital epinephrine administration, and the presence or absence of endotracheal intubation during ABGA at the hospital were not investigated. Second, even though the data mostly came from patients from 27 hospitals nationwide in South Korea from 2015 to 2022, the study population was limited since the incidence of pediatric OHCA is lower than that of adult cardiac arrest. Although the aORs for pH and PaCO_2_ in Table 4-B were relatively high at 4.15 and 3.99, respectively, the 95% confidence intervals were wide and the *p*-values were above 0.05. In Table 3, no patient had a shockable first monitored ECG or non-survival to hospital discharge. Therefore, we were unable to analyze the first monitored ECG variable by logistic regression. Third, among the KoCARC registry variables, the initial ABGA results were studied; however, it was difficult to collect artery blood only in cardiac arrest. The possibility of vein blood or mixed artery-vein blood cannot be completely ruled out. Fourth, it was difficult to determine the exact time from the patient’s arrival in the ED to the ABGA measurement. We collected results based on the variable “Initial ABGA in the emergency department,” making it challenging to analyze the timing after arrival and requiring caution in interpretation.

## 5. Conclusions

Our findings suggested that there was an association between PaO_2_ > 45.750 mmHg in initial ABGA and increased survival to hospital admission. Hence, PaO_2_ may serve as an independent prognostic value for survival to hospital admission in pediatric OHCA patients. However, few studies have investigated the association between the results of ABGA and the survival outcome of pediatric OHCA patients, so further large-scale studies are needed.

## Figures and Tables

**Figure 1 jpm-13-01061-f001:**
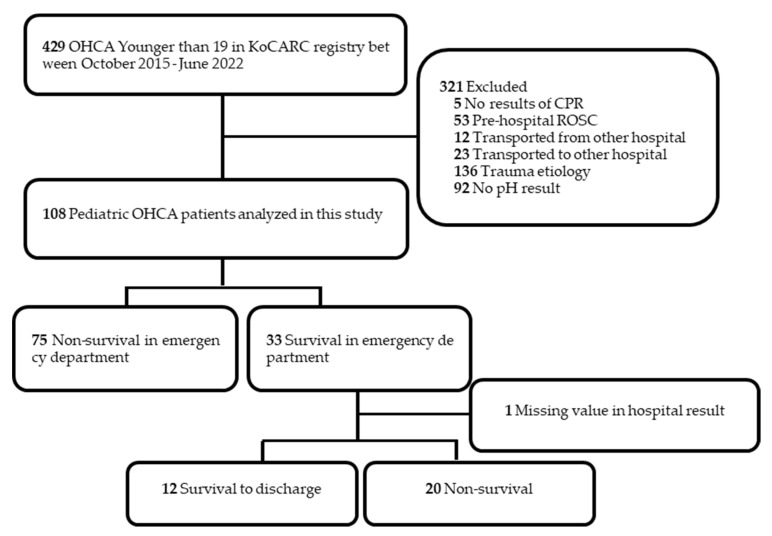
Flow diagram of the study. KoCARC, Korean Cardiac Arrest Resuscitation Consortium; ED, Emergency department; OHCA, Out-of-hospital cardiac arrest; ROSC, Return of spontaneous circulation; CPR, Cardiopulmonary resuscitation.

**Table 1 jpm-13-01061-t001:** Baseline characteristics of included patients with survival to hospital admission and non-survival in the emergency department.

	Survival to Hospital Admission		* *p*-Value
	Survival (*n* = 33)	Non-Survival (*n* = 75)	
Age (years)	0.0 [0.0; 6.0]	2.0 [0.0; 12.0]	0.146
Age category			0.595
<1 month	3 (9.1%)	3 (4.0%)	
1 to 12 months	13 (39.4%)	24 (32.0%)	
1 to 4 years	7 (21.2%)	14 (18.7%)	
5 to 12 years	5 (15.2%)	17 (22.7%)	
13 to 18 years	5 (15.2%)	17 (22.7%)	
Sex(male), n (%)	21 (63.6%)	44 (58.7%)	0.785
Comorbidities, n (%)	9 (27.3%)	40 (53.3%)	0.022 *
Comorbidities			0.126
Lung or airway	2 (22.2%)	2 (5.0%)	
Neurologic	2 (22.2%)	18 (45.0%)	
Congenital heart disease	4 (44.4%)	6 (15.0%)	
Genetic or metabolic	1 (11.1%)	6 (15.0%)	
Hematology or oncology	0 (0.0%)	3 (7.5%)	
Others or unknown ^†^	0 (0.0%)	5 (12.5%)	
Witnessed arrest, n (%)	20 (60.6%)	36 (48.0%)	0.318
Bystander CPR, n (%)	23 (69.7%)	47 (62.7%)	0.627
Prehospital advanced airway management, n (%)	13 (39.4%)	24 (32.0%)	0.599
First monitored ECG, shockable (%)	3 (90.9%)	4 (5.3%)	0.759
ABGA results			
pH	6.78 ± 0.18	6.70 ± 0.23	0.074
PaCO_2_, mmHg ^††^	84.34 ± 32.86	99.80 ± 43.78	0.079
PaO_2_, mmHg	49.0 [29.90; 88.40]	34.70 [19.80; 58.60]	0.063
Lactate, mmol/L ^††^	14.66 ± 4.83	15.18 ± 5.15	0.687

CPR, cardiopulmonary resuscitation; ECG, electrocardiography. ^†^ Others or unknowns include renal disease (*n* = 1), liver disease (*n* = 1), and unknown disease (*n* = 3). ^††^ Unavailable (missing) values were excluded from calculations: patient number calculated from PaCO_2_ (*n* = 101), patient number calculated from lactate (*n* = 69). * *p* < 0.05 is significant.

**Table 2 jpm-13-01061-t002:** Univariable and multivariable logistic regression of out-of-hospital cardiac arrest pediatric patients with survival events.

	Odds Ratio	95% CI	* *p*-Value
(A) Univariable analysis			
Age	0.95	0.89–1.02	0.154
Age category			
<1 month	Ref.		
1 to 12 months	0.54	0.10–3.08	0.489
1 to 4 years	0.50	0.08–3.14	0.460
5 to 12 years	0.29	0.04–1.94	0.203
13 to 18 years	0.29	0.04–1.94	0.203
Sex (Female)	0.81	0.35–1.89	0.627
Comorbidities (No)	3.05	1.25–7.42	0.014 *
Witnessed arrest (Yes)	1.67	0.73–3.83	0.229
Bystander CPR (Yes)	1.37	0.57–3.30	0.482
Prehospital advanced airway management (Yes)	1.38	0.59–3.23	0.457
First monitored ECG (Shockable)	1.78	0.37–8.42	0.470
pH > 6.795	2.13	0.93–4.90	0.074
PaCO_2_ ≤ 95.750 (mmHg) ^†^	2.34	0.98–5.59	0.055
PaO_2_ > 45.750 (mmHg)	2.71	1.17–6.30	0.020 *
Lactate ≤ 16.600 (mmHg) ^†^	0.65	0.24–1.77	0.393
	Adjusted odds ratio	95% CI	* *p*-Value
(B) Multivariable analysis			
Comorbidities (No)	3.03	1.22–7.53	0.017 *
PaO_2_ > 45.750 (mmHg)	2.69	1.13–6.42	0.026 *

CI, confidence interval. ^†^ Unavailable (missing) values were excluded from calculations: patient number calculated from PaCO_2_ (*n* = 101); patient number calculated from lactate (*n* = 69). * *p* < 0.05 is significant.

**Table 3 jpm-13-01061-t003:** Baseline characteristics of hospital admission patients with survival discharge and non-survival discharge.

	Survival to Discharge		* *p*-Value
	Survival (*n* = 12)	Non-Survival (*n* = 20)	
Age (years)	3.0 [0.0; 6.5]	0.0 [0.0; 1.0]	0.104
Age category			0.452
<1 month	1 (8.33%)	2 (10.0%)	
1 to 12 months	3 (25.0%)	10 (50.0%)	
1 to 4 years	3 (25.0%)	4 (20.0%)	
5 to 12 years	3 (25.0%)	1 (5.0%)	
13 to 18 years	2 (16.67%)	3 (15.0%)	
Sex (male), n (%)	9 (75.0%)	11 (55.0%)	0.451
Comorbidities, n (%)	5 (41.67%)	4 (20.0%)	0.361
Comorbidities			0.268
Lung or airway	2 (40.0%)	0 (0.0%)	
Neurologic	1 (20.0%)	1 (25.0%)	
Congenital heart disease	1 (20.0%)	3 (75.0%)	
Genetic or metabolic	1 (20.0%)	0 (0.0%)	
Witnessed arrest	8 (66.7%)	11 (55.0%)	0.780
Bystander CPR	8 (66.7%)	15 (75.0%)	0.919
Prehospital advanced airway management, n (%)	4 (33.33%)	8 (40.00%)	1.000
First monitored ECG, shockable (%)	3 (25.0%)	0 (0.0%)	0.085
ABGA results			
pH	6.87 ± 0.15	6.73 ± 0.18	0.030 *
PaCO_2_, mmHg ^†^	75.07 ± 37.97	87.66 ± 29.57	0.314
PaO_2_, mmHg	48.6 [29.10; 87.20]	48.35 [26.10; 106.25]	0.938
Lactate, mmol/L ^†^	15.56 ± 4.50	13.46 ± 5.03	0.320

CPR, cardiopulmonary resuscitation; ECG, electrocardiography. ^†^ Unavailable (missing) values were excluded from calculations: patient number calculated from PaCO_2_ (*n* = 31); patient number calculated from lactate (*n* = 22). * *p* < 0.05 is significant.

**Table 4 jpm-13-01061-t004:** Univariable and multivariable logistic regression of out-of-hospital cardiac arrest pediatric patients with survival to discharge.

	Odds Ratio	95% CI	* *p*-Value
(A) Univariable analysis			
Age	1.06	0.94–1.19	0.367
Age category			
<1 month	Ref		
1 to 12 months	0.6	0.04–9.16	0.713
1 to 4 years	1.5	0.09–25.39	0.779
5 to 12 years	6.00	0.22–162.53	0.287
13 to 18 years	1.33	0.07–26.62	0.851
Sex (female)	0.40	0.08–1.97	0.264
Comorbidities (No)	0.35	0.07–1.71	0.195
Witnessed arrest (yes)	1.64	0.37–7.25	0.517
Bystander CPR (Yes)	0.67	0.14–3.20	0.613
Prehospital advanced airway management (Yes)	0.75	0.17–3.35	0.706
pH > 6.775	7.5	1.29–43.69	0.025 *
PaCO_2_ ≤ 66.500 (mmHg) ^†^	6.8	1.23–37.50	0.028 *
PaO_2_ > 47.850 (mmHg)	1.40	0.33–5.93	0.648
Lactate ≤ 14.865 (mmol/L) ^†^	0.31	0.05–1.80	0.191
	Adjusted odds ratio	95% CI	*p*-Value
(B) Multivariable analysis			
pH > 6.775	4.15	0.62–27.97	0.144
PaCO_2_ ≤ 66.500 (mmHg)	3.99	0.63–25.23	0.141

CPR, cardiopulmonary resuscitation. There were no patients with a first monitored ECG non-shockable rhythm and survival to discharge, so the first ECG rhythm was excluded. ^†^ Unavailable (missing) values were excluded from calculations: patient number calculated from PaCO_2_ (*n* = 31), patient number calculated from lactate (*n* = 22). * *p* < 0.05 is significant.

## Data Availability

Not applicable.

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
