# Peer review of "Arterial Blood Gas Analysis for Survival Prediction in Pediatric Patients with Out-of-Hospital Cardiac Arrest"

_jpm, 2023, doi:10.3390/jpm13071061_

Round 1

Reviewer 1 Report

I thank the authors for submission of their very valuable manuscript, entitled "Arterial blood gas analysis for survival prediction in pediatric out-of-hospital cardiac arrest." in which they evaluate prognostic relevant parameters in pediatric OHCA regarding the two different endpoints "survival to admission" and "survival to discharge"

Decision-making, management, and therapy during OHCA are difficult. Relevant parameters in adult and also pediatric patients after OHCA are critically and controversially discussed over the last years. 

Therefore, the evaluation of valuable parameters, as the authors did, that further support therapy and decision-making in this time crucial situation is essential. 

Whereas the overall study cohort includes 108 patients, the patient cohort for evaluation of the second endpoint includes only 32 patients (12/20). The authors should check Figure 1, the labeling of the last patient groups might be wrong. 

Even though, there was a significant association of concomitant comorbidities and pAO2 regarding the endpoint "survival to admission", there was no significant correlation between any parameter, and in my opinion, the more relevant prognostic endpoint "survival to discharge".

Moreover, as the authors describe correctly, the fact that pAO2 could be measured from arterial blood, vein blood, or mixed-arterial-vein blood unfortunately further restricts the results. 

Nevertheless, the discussion of "new" relevant parameters during pediatric OHCA should be promoted to further optimize the outcomes of these patients. 

Reviewer 2 Report

Thank you for the opportunity to review this manuscript.

I have several suggestions for the authors.

Do the authors include EMS-witnessed cases? It would be better if the authors treated EMS-witnessed OHCA cases as a different population since their characteristics is different from layperson-witnessed OHCA cases.

Did authors analyze Non-traumatic OHCA? Cardiogenic OHCA? I understand the number of pediatric OHCA is limited, but its characteristics should be described.

Is the data of prehospital epinephrine administration available? If so, please put the numbers in the table.

The biggest point of this study method is that the timing for ABGA. All ABGAs were conducted within a certain period after hospital arrival? Otherwise, the analysis of the study results requires cautions.

Finally, the authors stated that there is a chance that some results of ABGA might have been influenced by possible venous blood gas samples. I don’t know the distribution of the study data, but it would be safe to exclude possible venous blood gas data. 

Round 2

Reviewer 2 Report

Thank you for the revision.